# A general large-scale synthesis approach for crystalline porous materials

Xiongli Liu[1], An Wang[2], Chunping Wang[2], Jinli Li[1], Zhiyuan Zhang[1], Abdullah M. Al-Enizi ✪[3], Ayman Nafady[3], Feng Shui[1], Zifeng You[1], Baiyan Li ✪[1] ✉, Yangbing Wen[2] ✉ & Shengqian Ma ✪[4] ✉

Crystalline porous materials such as covalent organic frameworks (COFs), metal-organic frameworks (MOFs) and porous organic cages (POCs) have been widely applied in various fields with outstanding performances. However, the lack of general and effective methodology for large-scale production limits their further industrial applications. In this work, we developed a general approach comprising high pressure homogenization (HPH), which can realize large-scale synthesis of crystalline porous materials including COFs, MOFs, and POCs under benign conditions. This universal strategy, as illustrated in the proof of principle studies, has prepared 4 COFs, 4 MOFs, and 2 POCs. It can circumvent some drawbacks of existing approaches including low yield, high energy consumption, low efficiency, weak mass/thermal transfer, tedious procedures, poor reproducibility, and high cost. On the basis of this approach, an industrial homogenizer can produce 0.96 ~ 580.48 ton of high-performance COFs, MOFs, and POCs per day, which is unachievable via other methods.

In past decades, covalent organic frameworks (COFs), metal-organic frameworks (MOFs), and porous organic cages (POCs) have been recognized as attractive class of crystalline porous materials owing to their unique features of high surface area, adjustable pore/window sizes, modular feature with tunable functional sites, high thermal/chemical stabilities[1–4]. And they have shown great potentials in the fields of gas storage/separation[5–7], sensing[8–10], electrochemical[11–13], energy storage[14–16], catalysis[17–19], pollutant treatment[20–22], etc. Therefore, sorts of synthetic techniques includes hydrothermal/solvothermal[17,23,24], ionothermal[25,26], solvent-free synthesis[27,28], sonochemical synthesis[29–31], microwave[32–34], plasma[35,36], microfluidic synthesis[37,38], microreactor synthesis[39], continuous flow synthesis[40,41], mechanochemical (MC) synthesis[42–45], and twin-screw extruder (TSE) approach[46–48] have been investigated and developed for the preparation of crystalline porous materials such as COFs, MOFs, and POCs. However, there are only a limited number of MOFs that were commercially produced[34], which may be attributed to that traditional synthesis methods have practical drawbacks including low yield, high energy consumption, low efficiency, notorious preparation setup, high operating costs, low product performance, weak mass/thermal transfer, and poor reproducibility. Moreover, the lack of continuous-production for all the existing approaches constitutes a significant barrier for the industrialization of these crystalline porous materials. Therefore, the development of general synthetic technologies for synthesizing these crystalline porous materials that can meet the essential requirements of practical industrial production with features of simple, rapid, continuous, high performance, large-scale, inexpensive, reproducible, and high yield remains a challenge[49].

High pressure homogenization (HPH) technology is a typical industrial productive approach in biological, pharmaceutical, food, chemical and other industrial polymer synthesis with the advantages of commercially available instrument with low-cost, simple operation at room temperature, low energy consumption, high production efficiency and continuous production[50]. Such approach can efficiently

[1]School of Materials Science and Engineering, National Institute for Advanced Materials, TKL of Metal and Molecule-Based Material Chemistry, School of Materials Science and Engineering & Smart Sensing Interdisciplinary Science Center, Nankai University, Tianjin 300350, P. R. China. [2]Tianjin Key Laboratory of Pulp and Paper, Tianjin University of Science and Technology, Tianjin 300457, P. R. China. [3]Department of Chemistry, College of Science, King Saud University, Riyadh 11451, Saudi Arabia. [4]Department of Chemistry, University of North Texas 1508 W Mulberry St, Denton, TX 76201, USA. ✉e-mail: libaiyan@nankai.edu.cn; yangbingwen@tust.edu.cn; Shengqian.Ma@unt.edu

disperse reactants in solvent which thus results in the high mass/thermal transfer efficiency in compared with conventional MC and TSE synthetic technology in synthesizing crystalline porous materials. Moreover, it can offer continuously synthesis via consecutive injection of the reactants, which exhibits the practical feasibility in industrial large-scale production of crystalline porous materials in contrast to the others traditional approaches such as hydrothermal/solvothermal, sonochemical synthesis, microwave and so on. Upon the above advantages, in this contribution, we developed for the first time a general approach based on HPH technology, which offers a continuous large-scale synthesis of crystalline porous materials including COFs, MOFs, and POCs with high-performance and high efficiency under benign conditions. The choice of HPH approach for highly efficient synthesis of these crystalline porous materials is mainly attributed to the following reasons: (i) cavitation of the pipes occurs during homogenization, which possibly results in local vacuum that could prevent the partial oxidation of organic ligands, thereby enhancing the reaction process[50], (ii) the solution media under HPH conditions facilitate mass transfer of reactants and hence increasing the reproductive as well as the amount of conjugated units compared with conventional MC and TSE methods[42,46], (iii) the mechanical force such as shear stress, collision, high frequency shock, and turbulent flow will dramatically speed up the formation of COFs, MOFs, and POCs[51], (iv) the HPH method could achieve the continuous bulk production of COFs, MOFs, and POCs via the consecutive injection of the reactants in solution media[52] (Fig. 1). Therefore, high pressure homogenization (HPH) approach can not only overcome shortcomings of low yield, high energy consumption, low efficiency, and sophisticated preparation process for conventional methods such as hydrothermal/solvothermal, ionothermal, microwave, and sonochemical synthesis etc., but also circumvent the weaknesses of low crystallinity, poor mass/thermal transfer, poor reproducibility, and discontinuously production for reported MC and TSE approaches.

## Results

To illustrate our strategy, we conducted a proof of concept study using 4 COFs (TpPa-1, TpPa-2, TpBD, DAAQ), 4 MOFs (HKUST-1, NH$_2$-MIL-53(Al), ZIF-8, and ZIF-67), and 2 POCs (CPOC-301, and CC3R-OH) (Supplementary Fig. 1). As for COFs, a mixture of water, acetic acid, 1,3,5-triformylphloroglucinol (Tp) and either phenylenediamine (Pa-1) (for HPH-TpPa-1), 2,5-dimethyl-p-phenylenediamine (Pa-2) (for HPH-TpPa-2), benzidine (BD) (for HPH-TpBD), or 2,6-diaminoanthraquinone (DAAQ) (for HPH-DAAQ) were placed in a beaker and stir for 5 min; the suspension was pumped into the high pressure homogenizer and homogenized for 5, 15, 30, or 60 min, and the light-yellow powder changed to orange during the homogenization process, and finally a red powder was obtained, indicating the occurrence of polymerization reactions with increased amount of conjugated units (Supplementary Fig. 2). A similar synthesis procedure was carried out for MOFs and POCs, the details was shown in supporting information (Supplementary Figs. 3–8). The solid-state UV−vis spectra studies show that the peak intensity of the products increases with homogenization time during the HPH progress, and the adsorption wavelengths of samples obtained by HPH approach are consistent with those of solvothermally synthesized countparts, which further indicates the occurrence of polymerization or coordination reactions (Supplementary Figs. 9–11). The facile formation of these crystalline porous materials could be ascribed to the shear stress, collision and cavitation interactions during the homogenization process, and the products obtained based on HPH method were named according to the different homogenization times.

Powder x-ray diffraction (PXRD) of HPH-TpPa-1, HPH-TpPa-2, HPH-TpBD, HPH-DAAQ, HPH-HKUST-1, HPH-ZIF-8, HPH-ZIF-67, HPH-NH$_2$-MIL-53(Al), HPH-CC3R-OH, and HPH-CPOC-301 exhibited strong 2θ peaks at low angles (Fig. 2), indicative of good crystallinity for the samples obtained by HPH method. All of the observed PXRD patterns for HPH-COFs, HPH-MOFs, and HPH-POCs matched well with the

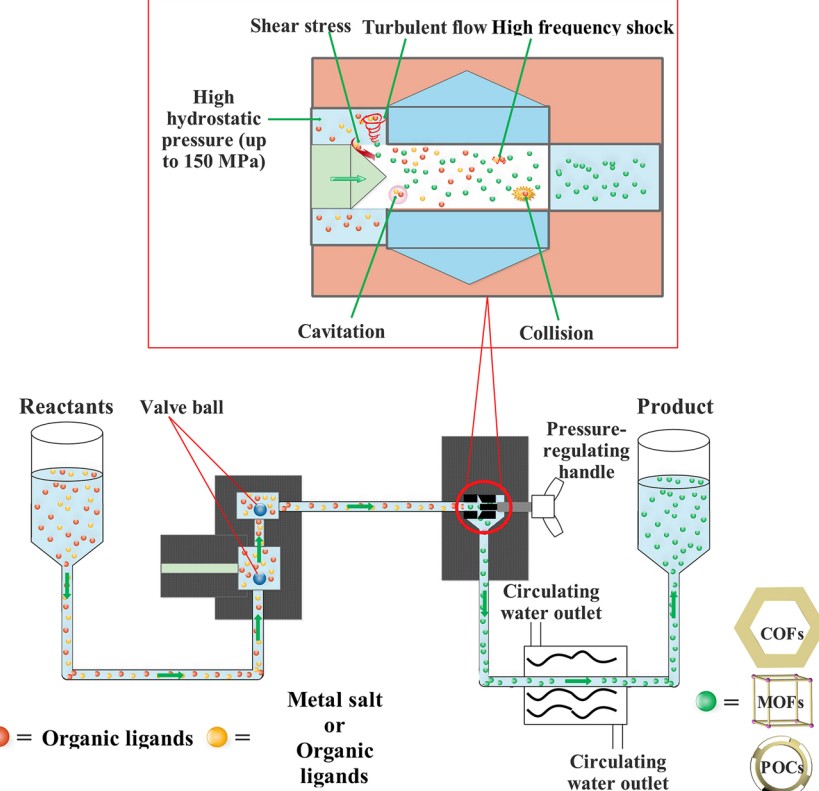

**Fig. 1 | High pressure homogenization strategy for synthesizing crystalline porous materials.** Schematic illustration of the general synthesis process of crystalline porous materials including COFs, MOFs, and POCs using high pressure homogenization strategy.

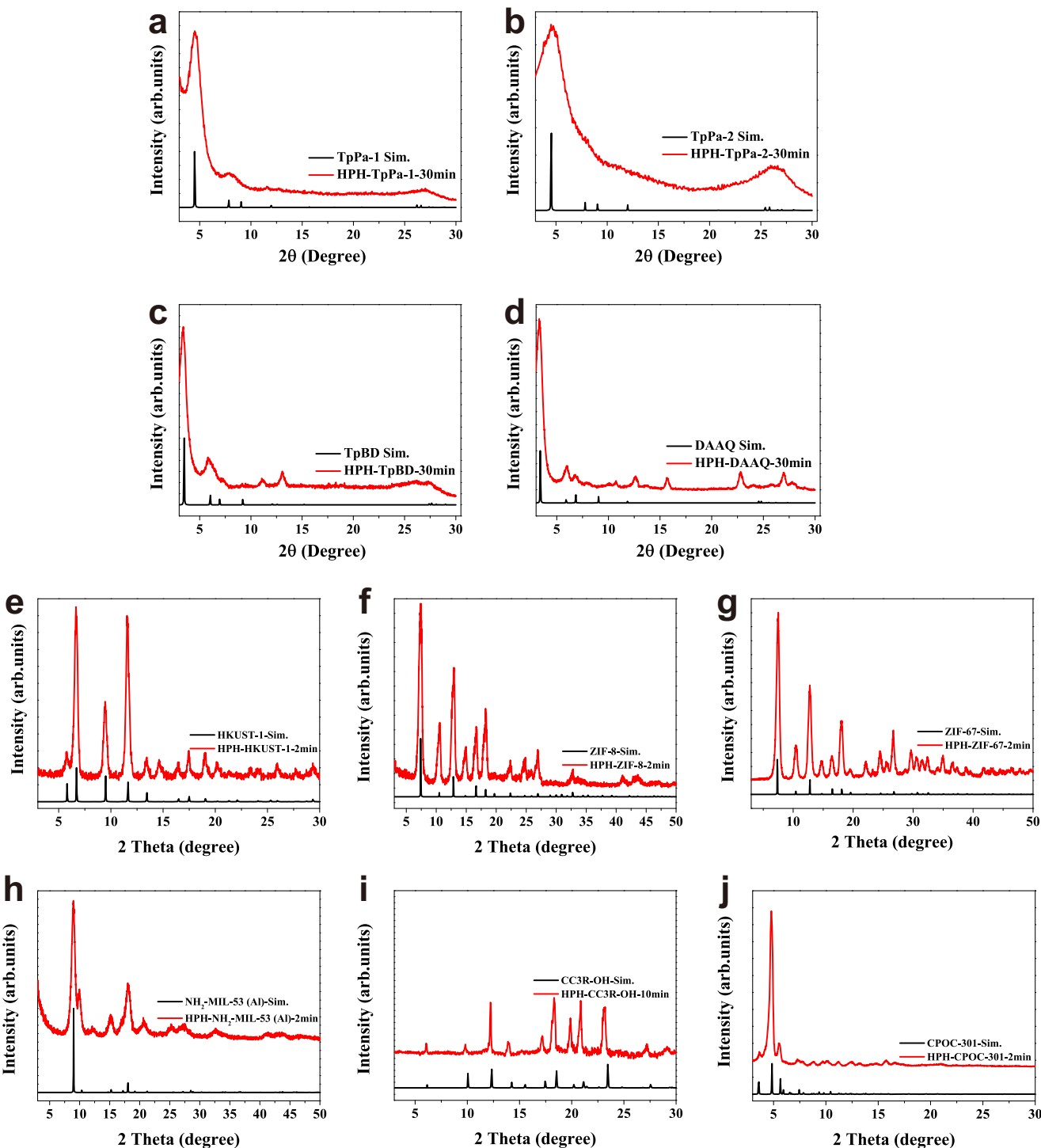

**Fig. 2 | PXRD patterns for crystalline porous materials synthesized via high pressure homogenization (red) and simulated curves (black). a** HPH-TpPa-1-30min, **b** HPH-TpPa-2-30min, **c** HPH-TpBD-30min, **d** HPH-DAAQ-30min, **e** HPH-HKUST-1-2min, **f** HPH-ZIF-8-2min, **g** HPH-ZIF-67-2min, **h** HPH-NH₂-MIL-53(Al) −2min, **i** HPH-CC3R-OH-10min, and **j** HPH-CPOC-301-2min.

simulated patterns and Pawley refinements (Fig. 2 and Supplementary Fig. 12). Scanning electron microscopy (SEM) images indicated that the obtained HPH-TpPa-1-30min, HPH-TpPa-2-30min, HPH-TpBD-30min, HPH-DAAQ-30min, HPH-ZIF-8-2min, HPH-ZIF-67-2min, HPH-HKUST-1-2min, HPH-NH₂-MIL-53(Al)-2min, HPH-CPOC-301-2min, and HPH-CC3R-OH-10min were stacked to plate-like, blocky, nets, rod-shaped, hexagon, rectangle, diamond, flower-like, massive, and octahedron particles, respectively. The SEM images of the obtained HPH-COFs, HPH-MOFs, and HPH-POCs are also compared with their solvothermal

analogues. The samples obtained via solvothermal method exhibit a larger size as compared to HPH products, which could be ascribed to the longer crystallization time of the solvothermal method (Supplementary Figs. 13 and 14).

**Synthesis of COFs via HPH approach**

To gain better insight to the formation of the HPH-COFs, we firstly investigated the effect of homogenization time on the reaction. The formation of HPH-TpPa-1 was monitored by PXRD and Fourier

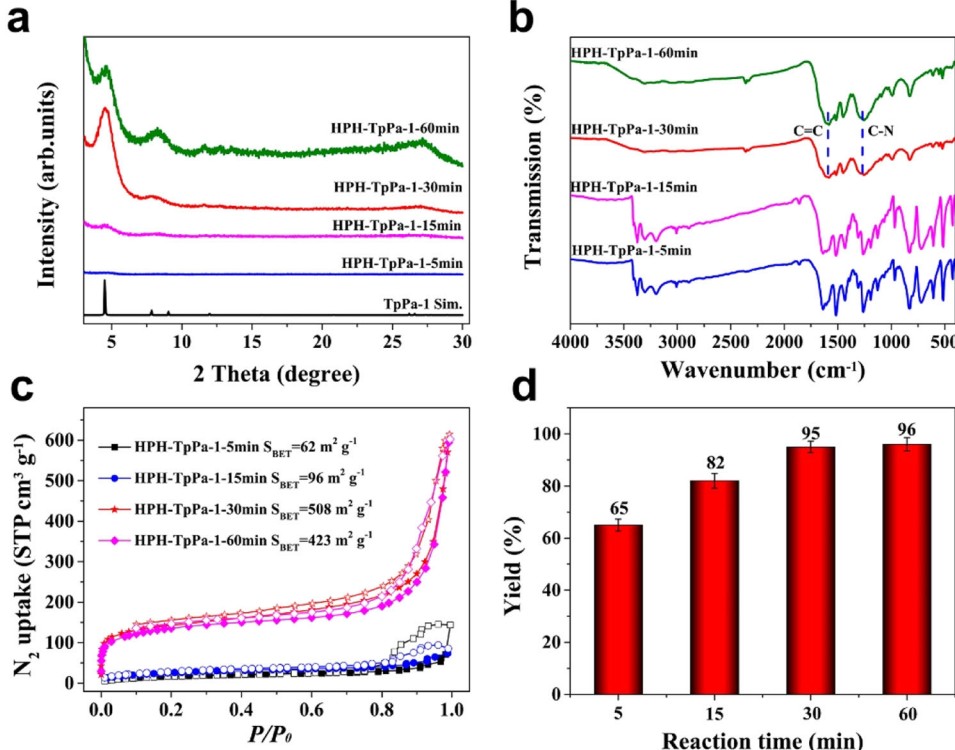

**Fig. 3 | Systematic investigation of the effect of reaction time on the TpPa-1 formation by high pressure homogenization (HPH) approach. a** PXRD patterns of HPH-TpPa-1 at different reaction times. **b** FT-IR spectra of HPH-TpPa-1 at different reaction times. **c** N$_2$ sorption isotherms of HPH-TpPa-1 at different reaction times. **d** The yields of HPH-TpPa-1 at different reaction times. All the error bars in this figure represent the standard deviation ($n$ = 3 independent experiments), data are presented as mean values ± SD.

transform infrared (FT-IR) spectroscopy. Figure 3a shows that the crystallinity of HPH-TpPa-1 was enhanced from 5 to 30 min and then a little weakened from 30 to 60 min. The slight decrease in crystallinity of HPH-TpPa-1 after a long time homogenization may be due to possible mechanical exfoliation of 2D layers in forming COF. In addition, the disappearance of the N−H stretching frequency (3100–3300 cm$^{-1}$, corresponding to the free amine groups, Fig. 3b) and the appearance of C−N peaks (1250 cm$^{-1}$) and C=C (1578 cm$^{-1}$) peaks also confirmed the successful formation of HPH-TpPa-1, which is consistent with reported solvothermal counterparts[42,53,54]. Also, the successfully synthesis of HPH-TpPa-1 was verified by the consistence in $^{13}$C cross-polarization magic-angle-spinning (CP-MAS) solid-state NMR spectroscopy (Supplementary Fig. 15). The permanent porosity of HPH-TpPa-1 was evaluated by nitrogen adsorption isotherms measured at 77 K. The results showed that the prolong of homogenization times led to a large enhance of BET surface area from 62 m$^2$ g$^{-1}$ (5 min) to 508 m$^2$ g$^{-1}$ (30 min) in synthesizing HPH-TpPa-1 (Fig. 3c). And further extension of homogenization time from 30 to 60 min resulted in a slight decline in product quality, in consistence with PXRD results, which might be ascribed to the possible exfoliation of forming COFs by continuous mechanical force. Yields of HPH-TpPa-1 was increased from 65% for 5 min to 95% for 30 min homogenization, and longer homogenization times (60 min) showed negligible change in yield (Fig. 3d). Thus, 30 min is the optimal homogenization time for the formation of HPH-TpPa-1, which is 144 times faster than the conventional solvothermal method (72 h)[54]. To demonstrate the generality of HPH strategy for COF synthesis, three other COFs including TpPa-2, TpBD, and DAAQ[42,53] were also synthesized by HPH method (Supplementary Figs. 16–19), which thus highlight the significant advantages of HPH approach. In addition to homogenization time, homogenization pressure is also a critical parameter for controlling the quality of products. The results showed that 100 MPa is an optimum pressure for

producing HPH-COFs (Supplementary Figs. 20–23). Moreover, control experiments illustrate that the conventional room-temperature synthesis can not produce these COF materials after stirring for 30 min (Supplementary Figs. 24–27, Supplementary Table 1), which thus highlights the significant advantages of HPH approach.

Nitrogen sorption isotherms collected at 77 K were used to quantify the performance of the HPH-COFs (Fig. 4a). The BET surface areas of HPH-TpPa-1, HPH-TpPa-2, HPH-TpBD, and HPH-DAAQ were determined to be 508, 312, 542, and 335 m$^2$ g$^{-1}$, which are significantly higher than those of mechanochemical (MC) COFs with a surface area of 61, 56, 35, and 43 m$^2$ g$^{-1}$ for TpPa-1 (MC), TpPa-2 (MC), TpBD (MC), and DAAQ (MC) (Fig. 4b), respectively[42,53]. The hysteresis for the N$_2$ sorption isotherms of HPH-COFs may be ascribed to capillary condensation[46]. It is worth noting that the BET surface areas (Fig. 4b) and pore size distributions (Supplementary Fig. 28) of HPH-COFs are comparable to those of solvothermally synthesized COFs[42,53] (Fig. 4b). In addition, thermogravimetric analysis (TGA) profiles indicated that all HPH-COFs exhibiting high thermal stabilities up to ~350 °C (Supplementary Fig. 29). Also, these HPH-COFs exhibit good chemical stabilities, consistent with the reported results (Supplementary Figs. 30 and 31). In addition, the power consumption for preparing HPH-COFs is 0.36 KWh as measured by electric power monitor (homogenized for 30 min under 100 MPa), which is about 61 times lower than the consumption during the preparation of ST-TpPa-1, ST-TpPa-2, ST-TpBD (22.23 KWh, 120 °C, 72 h under oven) and 40 times lower than the consumption in the synthesis of ST-DAAQ (14.82 KWh, 120 °C, 48 h under oven)[42,53] (Supplementary Fig. 32). Hence, these results highlighted the advantages of HPH method in synthesizing high quality COFs. Moreover, despite twin-screw extruder (TSE) approach was reported to be able to realize a large-scale production of COFs with high surface area[46]. However, the essential problems associated with TSE approach including poor reproducibility and uniformity of

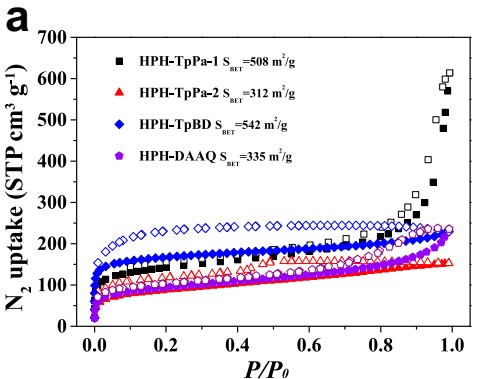
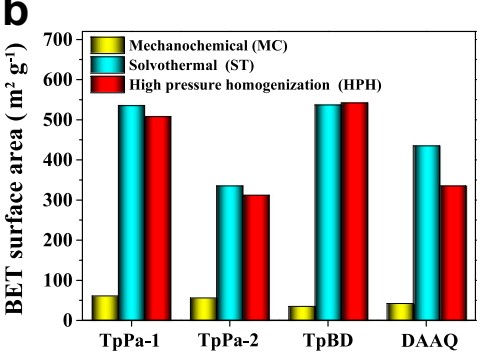

**Fig. 4 | The comparison of BET surface areas of COFs by using different synthetic methods. a** N$_2$ sorption isotherms of COFs synthesized using the high pressure homogenization approach. **b** The comparisons of BET surface areas of COFs synthesized via mechanochemical (MC, yellow), solvothermal (ST, cyan), and high pressure homogenization (HPH, red) approach.

product because of mass/thermal transfer obstacle, largely hindered its practical industrial production.

On the contrary, HPH method can achieve the continuous bulk production of COFs by consecutively injecting the reactants suspension into the homogenizer, bringing large-scale industrial production of COFs into practice. In this regard, we can easily obtained 54.6 g HPH-TpPa-1 within 30 min by using a small size industrial homogenizer (discharge: 180 L h⁻¹) (Supplementary Fig. 33), which thus highlight the industrialization potential of HPH approach. Furthermore, the solvent and catalyst could be readily recycled and reused without the loss of ability in synthesizing next batch of COF materials. This can be verified by the appearance of similar PXRD peaks (Supplementary Fig. 34) and the retention of BET surface areas of HPH-TpPa-1 (Supplementary Fig. 35) after the solvent and catalyst reused for every six times by adding 10 mL acetic acid. These results thus indicated the well reproducibility of HPH method due to its highly mass/thermal transfer in contrast with TSE and MC approach. Accordingly, the Space-time-yield (STY) of 4 COFs based on HPH method was calculated to $1.517 \times 10^3 \sim 2.381 \times 10^3$ kg m⁻³ day⁻¹ and it can produce HPH-COFs with production rate of 0.96~1.44 ton/homogenizer/day when using industrial homogenizer (discharge, 10000 L h⁻¹), which can greatly meet most industrial needs (Supplementary Table 2). In a word, HPH approach is undoubtedly the low-cost and high-efficiency methods that can realize large-scale synthesis of high-performance COF materials with high mass/thermal transfer as well as high reproducibility.

## Synthesis of MOFs via HPH approach

To demonstrate the generality of HPH approach for synthesizing other types of crystalline porous materials, as exemplified by MOFs, we selected four well-known MOFs including HKUST-1, ZIF-8, ZIF-67, and NH$_2$-MIL-53(Al) as proof of principle[55–58]. Taking HKUST-1 as an example, as shown in Fig. 5a, b, the successful synthesis of HKUST-1 by HPH approach was confirmed by PXRD and FT-IR studies. PXRD analysis indicated the disappear of peak for copper source (Cu(OH)$_2$, $2\theta = 23.7°$)[59] and the appearance of consistent peaks as simulated HKUST-1 after homogenizing for 1 min, which suggested the high efficiency of HPH approach for MOFs synthesis (Fig. 5a, Supplementary Fig. 36). The FT-IR spectra of HPH-HKUST-1 have similar characteristic peaks including asymmetric and symmetric stretching of the carboxylate group (located at 1644, 1447, and 1370 cm⁻¹) as well as stretching vibration band of Cu−O (centered at 728 and 489 cm⁻¹), comparable to the reported results[59], indicating the successful formation of HKUST-1 (Fig. 5b). The BET surface area of HPH-HKUST-1 can reach an optimized maximum of 1725 m² g⁻¹ (Fig. 5c) in 2 min homogenization with a yield of 98% (Fig. 5d). All the above-mentioned results indicated 2 min homogenization is an optimized condition for HKUST-1 synthesis. In

addition, three other MOFs including ZIF-8, ZIF-67, and NH$_2$-MIL-53(Al) can be also successful prepared by HPH technology, which further suggests the versatility and generality of HPH approach towards the synthesis of crystalline porous materials (Supplementary Figs. 37–40). It is worth mentioning that a large amount of excess of 2-methylimidazole is needed during the synthesis ZIF-8 and ZIF-67, and the optimum stoichiometric ratios of Zn²⁺:2-methylimidazole for ZIF-8 and Co²⁺:2-methylimidazole for ZIF-67 are 1:40 and 1:35, respectively (Supplementary Figs. 41 and 42). In addition, the optimal homogenization pressure is 100 MPa for HPH-MOFs (Supplementary Figs. 43–46). Control experiments of preparing the related MOFs via conventional room temperature synthesis (after stirring for 2 min) can only afford MOFs with poor crystallinity and low surface areas under the same reaction time (Supplementary Figs. 47–50, Supplementary Table 1), which thus highlights the significant advantages of HPH approach. Moreover, we employed water instead of organic solvents as environment-friendly solvent for preparing HKUST-1 and ZIF-67. Unfortunately, HKUST-1 can not be obtained using water as the solvent (Supplementary Fig. 51, Supplementary Table 3). Moreover, the optimized BET surface area (BET = 613 m² g⁻¹) of ZIF-67 by using water as the solvent was much lower than that of using methanol as the solvent (BET = 1282 m² g⁻¹) (Supplementary Fig. 52, Supplementary Table 3). These results indicate that the presence of organic solvents is beneficial for forming high-quality HKUST-1 and ZIF-67.

The BET surface areas of HPH-HKUST-1-2min, HPH-ZIF-8-2min, HPH-ZIF-67-2min, and HPH-NH$_2$-MIL-53(Al)−2min were determined to be 1725, 1331, 1282, and 716 m² g⁻¹, respectively (Fig. 6a), which are comparable to the reported sovlothermally prepared counterparts[55–58] (Fig. 6b, Supplementary Table 4). In addition, HPH-MOFs exhibit high thermal and chemical stabilities, which are comparable to the reported counterparts[60–63] (Supplementary Figs. 53 and 54). Moreover, the power consumption for preparing HPH-HKUST-1-2min, HPH-ZIF-8-2min, HPH-ZIF-67-2min, HPH-NH$_2$-MIL-53(Al)-2min is 0.024 KWh as measured by electric power monitor (homogenized for 2 min under 100 MPa), which is about 81, 1128, 869, and 1116 times lower than the ST-HKUST-1 (1.97 KWh, 85 °C, 8 h under oven)[55], ST-ZIF-8 (27.07 KWh, 140 °C, 48 h under oven)[56], ST-ZIF-67 (20.86 KWh, 100 °C, 72 h under oven)[57], and ST-NH$_2$-MIL-53(Al) (26.77 KWh, 130 °C, 72 h under oven)[58], respectively (Supplementary Fig. 55). Those results thus highlighted the advantages of HPH method in synthesizing high quality MOFs.

Similarity to COFs, HPH method can achieve the bulk production of MOFs. In this context, we obtained 188.9 g high-performance HPH-HKUST-1 in only 2 min by using a small size industrial homogenizer (discharge: 180 L h⁻¹) (Supplementary Figs. 56 and 57), which thus highlight the industrialization potential of the HPH technology for MOF synthesis. The STY of HPH-HKUST-1 ($4.4877 \times 10^4$ kg m⁻³ day⁻¹)

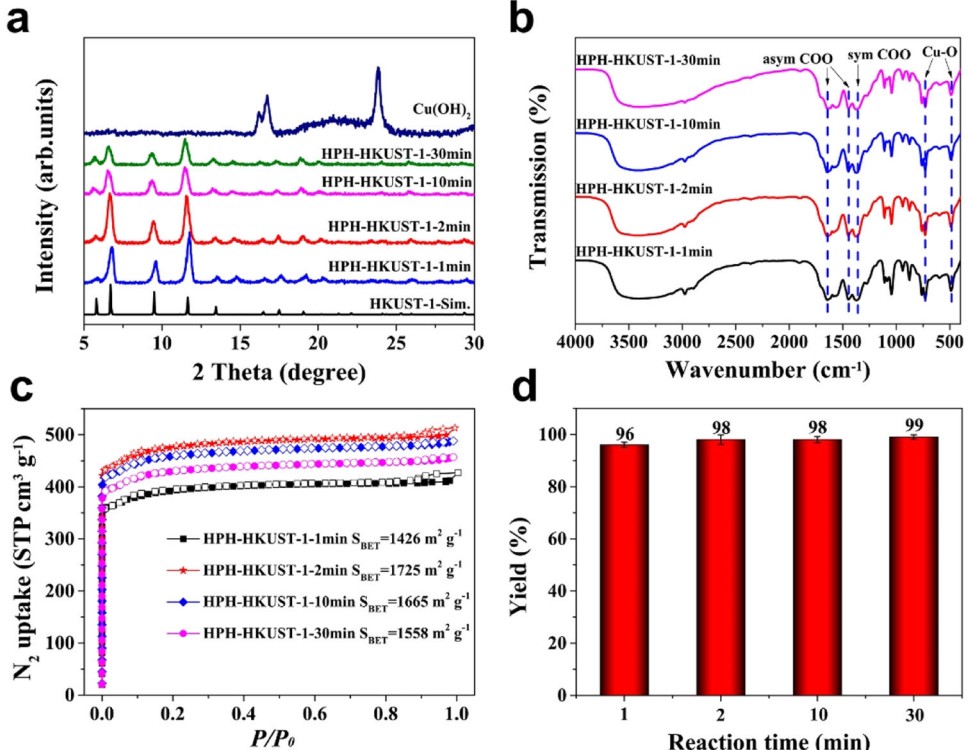

**Fig. 5 | Systematic investigation of the effect of reaction time on the HKUST-1 formation by high pressure homogenization (HPH) approach. a** PXRD patterns of HPH-HKUST-1 at different reaction times. **b** FT-IR spectra of HPH-HKUST-1 at different reaction times. **c** $N_2$ sorption isotherms of HPH-HKUST-1 at different reaction times. **d** The yields of HPH-HKUST-1 at different reaction times. All the error bars in this figure represent the standard deviation ($n = 3$ independent experiments), data are presented as mean values ± SD.

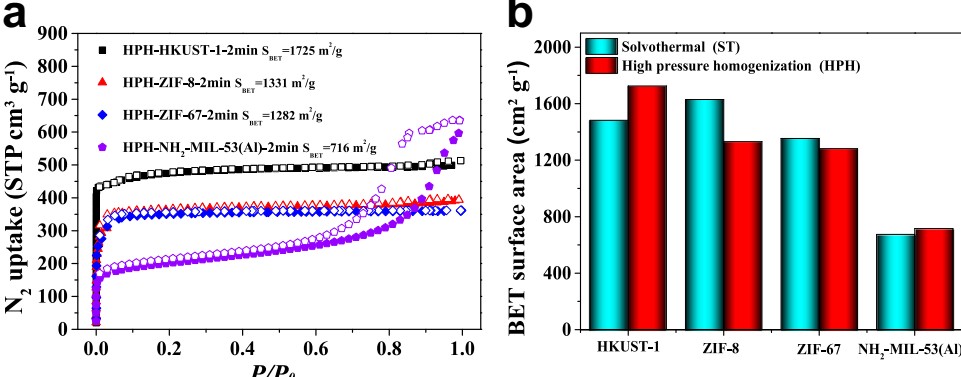

**Fig. 6 | The comparison of BET surface areas of MOFs by using different synthetic methods. a** $N_2$ sorption isotherms of MOFs synthesized by using HPH approach. **b** The comparison of the BET surface areas of MOFs synthesized via solvothermal (ST) method (cyan), and HPH approach (red).

is 199 times higher than that of producing the commercial HKUST-1 in BASF (Badische Anilin-und-Soda-Fabrik, trade name: Basolite C 300, 225 kg m$^{-3}$ day$^{-1}$)[59] (Supplementary Table 4), which suggests the high industrial potential of HPH technology. Furthermore, the solvent could be readily recycled and reused without the loss of ability in synthesizing next batch of HPH-HKUST-1 (Supplementary Figs. 58 and 59). Accordingly, the STY of synthesizing other three MOFs based on HPH method was calculated to $1.2344 \times 10^4 \sim 5.8048 \times 10^4$ kg m$^{-3}$ day$^{-1}$ (Supplementary Table 2), and it can produce MOFs with a production rate of 123.44 ~ 580.48 ton/homogenizer/day when using industrial homogenizer (discharge, 10,000 L h$^{-1}$) (Supplementary Table 2), suggesting the great potential in industrial production of MOFs.

## Synthesis of POCs via HPH approach

In addition to COFs and MOFs, another kind of crystalline porous materials, POCs, can also be synthesized via HPH approach. Take CPOC-301[24] as an example, PXRD studies indicated the appearance of consistent peaks as simulated CPOC-301 after homogenizing for 1 min, which suggested the successful synthesis of HPH-CPOC-301 by HPH approach (Fig. 7a). In addition, FT-IR spectra of HPH-CPOC-301 have similar characteristic peaks consistence with its solvothermal counterparts[24] including C−N (1279 cm$^{-1}$), C = O (1628 cm$^{-1}$), C = C (1579 cm$^{-1}$), and C = C−H (2955 cm$^{-1}$), indicating the successful formation of CPOC-301 (Fig. 7b). The BET surface area of HPH-CPOC-301 can reach an optimized maximum value of 1988 m$^2$ g$^{-1}$ (Fig. 7c) after 2 min homogenization with a yield of 90% (Fig. 7d). CC3R-OH[64] could also be

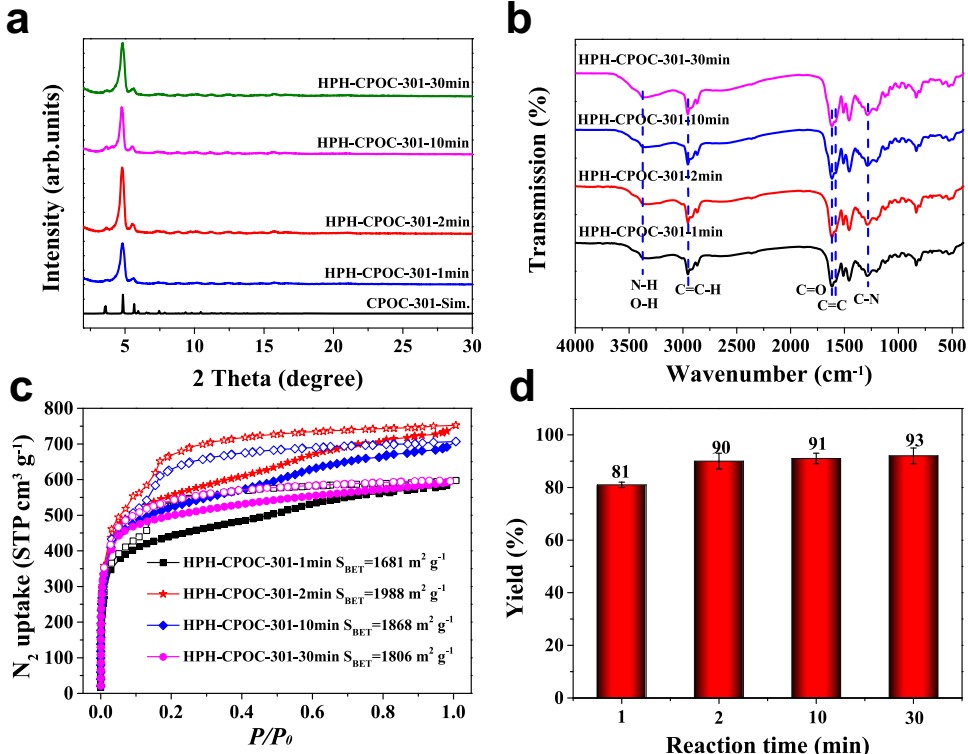

**Fig. 7 | Systematic investigation of the effect of reaction time on the CPOC-301 formation by high pressure homogenization (HPH) approach. a** PXRD patterns of HPH-CPOC-301 at different reaction times. **b** FT-IR spectra of HPH-CPOC-301 at different reaction times. **c** $N_2$ sorption isotherms of HPH-CPOC-301 at different reaction times. **d** The yields of HPH-CPOC-301 at different reaction times. All the error bars in this figure represent the standard deviation ($n = 3$ independent experiments), data are presented as mean values ± SD.

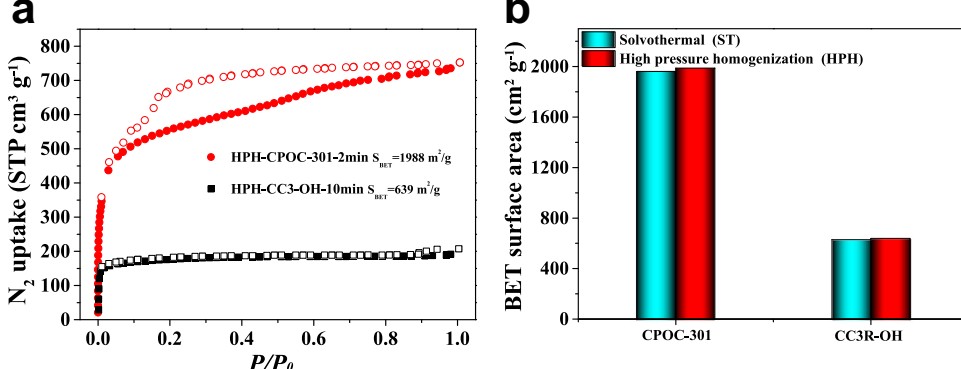

**Fig. 8 | The comparison of BET surface areas of POCs by using different synthetic methods. a** $N_2$ sorption isotherms of POCs synthesized by using HPH approach. **b** The comparison of BET surface areas of POCs synthesized via solvothermal (cyan), and HPH approach (red).

successful prepared by HPH approach, which further verifies the adaptability of HPH approach in POCs synthesis (Supplementary Fig. 60). The optimal homogenization pressure is 100 MPa for HPH-POCs synthesis (Supplementary Figs. 61 and 62). Control experiment indicated that the conventional room temperature synthesis via regular stirring cannot produce these POC materials (Supplementary Figs. 63 and 64, Supplementary Table 1), which thus highlights the advantages of HPH approach. The attempt to replace the organic solvent by using the environment friendly water to prepare both CPOC-301 and CC3R-OH is not successful, as indicated by the different PXRD patterns and the loss of BET surface area in water system under the same HPH conditions (Supplementary Figs. 65 and 66, Supplementary Table 3).

The BET surface areas of HPH-CPOC-301-2min and HPH-CC3R-OH-10min were determined to be 1988, and 639 $m^2$ $g^{-1}$ (Fig. 8a) respectively, which are comparable to the reported solvothermal methods (Fig. 8b)[24,64]. In addition, HPH-POCs exhibit high thermal and chemical stabilities, which are consistent with the reported solvothermal counterparts (Supplementary Figs. 67 and 68)[24,64]. Moreover, the power consumption of synthesizing HPH-CPOC-301-2min, and HPH-CC3R-OH-10min is 0.024 and 0.11 KWh, respectively (homogenized for 2 and 10 min under 100 MPa), which is about 325, and 4 times lower than that of ST-CPOC-301 (7.82 KWh, 65 °C, 48 h under oven)[24], and ST-CC3R-OH (0.54 KWh, 100 °C, 4 h for magnetic stirrer)[64], respectively (Supplementary Fig. 69). These results thus highlights the advantages of HPH approach for synthesizing high-performance POCs.

Moreover, HPH-POCs can also be produced in a large-scale, which can produce 53.2 g HPH-CPOC-301 within 2 min by using a small size industrial homogenizer (discharge: 180 L h⁻¹) (Supplementary Fig. 70). Accordingly, the Space-time-yield (STY) of two POCs based on HPH method was calculated to $3.828 \times 10^3 \sim 2.0821 \times 10^4$ kg m⁻³ day⁻¹ and it can produce CPOC-301 and CC3R-OH with production rate of 38.28~208.21 ton/homogenizer/day when using industrial homogenizer (discharge, 10,000 L h⁻¹) (Supplementary Table 2), which can greatly satisfy the requirements of industrial production. Furthermore, the solvent could be readily recycled and reused without the loss of ability in synthesizing next batch of HPH-CPOC-301. (Supplementary Figs. 71 and 72). Such approach undoubtedly exhibits the great advantages for POCs synthesis compared with other conventional methods.

## Discussion

In conclusion, we developed herein a high pressure homogenization approach to prepare high performance crystalline porous materials including COFs, MOFs, and POCs with advantageous features of simple, rapid, continuous, large-scale, and low cost. HPH method not only can overcome the shortcomings of low yield, high energy consumption, low efficiency, and sophisticated preparation process for conventional methods such as hydrothermal/solvothermal, ionothermal, microwave, sonochemical synthesis and so on, but also can circumvent the intrinsic drawbacks of poor mass/thermal transfer, poor reproducibility, and low production rate for the reported MC, TSE, microfluidic (MF), and microreactor (MR) methods (Supplementary Table 5). Moreover, HPH strategy provides facile access to continuous production of COFs, MOFs, and COFs with an extremely high production rate (0.96 ~ 580.48 ton day⁻¹) when using industrial homogenizer (discharge, 10,000 L h⁻¹) with ultrahigh space time yields ($1.517 \times 10^3 \sim 5.8048 \times 10^4$ kg m⁻³ day⁻¹). Practically, the advantageous features such as simple procedure, continuous process, large-scale, high reproducibility, low cost, and high efficiency for HPH methodology would bring industrial production of high performance crystalline porous materials (COFs, MOFs and POCs) into practice. Finally, we believe that the facile synthetic protocol with features of low cost, and high efficiency may replenish the existing synthesis processes and provide a new strategy for large-scale synthesis of a wide range of crystalline porous materials.

## Methods

### Materials and measurements

All reagents and solvents were commercially available and used as received. High pressure homogenization experiment was carried out using high pressure homogenizer (AH-PILOT, for Laboratory synthesis, 20 L h⁻¹; GYB180-18D, for large-scale synthesis, discharge, 180 L h⁻¹). Powder X-ray diffraction (PXRD) patterns were recorded on a Phillips PA Nalytical diffractometer with Cu Ka radiation (l = 1.5406 Å) by using a scan speed of 1° min⁻¹ and a step size of 0.02° in 2 Theta. Infrared spectra were recorded on a TENSOR 37 FT-IR spectrometer equipped with an attenuated total reflectance accessory. Thermogravimetric (TGA) analysis was performed on a TA Q500 thermogravimetric analyzer with a heating range of 20−800 °C using a 10 °C/min ramp under N₂. Solid state ¹³C cross polarization (CP) spectra were collected on a 11 Tesla magnet at a ¹³C frequency of 125.7 MHz under 12 kHz magic-angle spinning (MAS) conditions using Avance III WB 400. Scanning electron microscopy (SEM) images were obtained using a field emission SEM (JSM-7800F, imaged at 5 keV and 12 μA). Gas adsorption measurements were performed with Micromeritics ASAP 2020 plus.

### Synthetic procedures for HPH-COFs

The synthesis of TpPa-1, TpPa-2, TpBD, and DAAQ via high pressure homogenization (Laboratory synthesis, discharge: 20 L h⁻¹): 1,3,5-tri-formylphloroglucinol (Tp, 98%, 630 mg, 3 mmol) and corresponding amine: p-phenylenediamine (Pa-1, 99%, 480 mg, 4.5 mmol); 2,5-dime-thyl-p-phenylenediamine (Pa-2, 98%, 610 mg, 4.5 mmol); Benzidine (BD, 98%, 830 mg, 4.5 mmol) and 2,6-diaminoanthraquinone (AQ, 98%, 1020 mg, 4.5 mmol) were put in water-acetic acid medium (water, 300 mL, acetic acid, 99.5%, 150 mL) to form pre-synthetic slurries, respectively. The mixture was stirred for 5 min to achieve a homogenous suspension and then it was pumped into homogenizer and homogenized for different time intervals under 100 MPa. At different time intervals, the COF powders were collected, filtered, and washed with water, followed by ethanol (99.5%) for 3 times and finally dried under vacuum at 150 °C for 12 h.

### Synthetic procedures for HPH-HKUST-1

The synthesis of HPH-HKUST-1 was conducted in a high pressure homogenizer (Laboratory synthesis, discharge: 20 L h⁻¹). Typically, trimesic acid (99%, 12.17 g, 0.058 mol) was dissolved in ethanol (99.5%, 0.2 L). Then a suspension of Cu(OH)₂ (99%, 9.07 g, 0.093 mol) in water (0.1 L) was added to achieve a mixture. The mixture was pumped into homogenizer and homogenized under 100 MPa for 1, 2, 10, 30 min, respectively. At the time intervals, the blue MOF powders were collected, filtered, and washed with water and ethanol (99.5%) for 3 times. The product was finally dried under vacuum at 120 °C for 12 h.

### Synthetic procedures for HPH-CPOC-301

The synthesis of CPOC-301 was performed in a high pressure homogenizer (Laboratory synthesis, discharge: 20 L h⁻¹). C4RACHO (99%, 1640 mg, 2 mmol), p-Phenylenediamine (99%, 432 mg, 4 mmol), and mesitylene (99%, 60 mL) was added into a 250 mL beaker and stir for 5 min. The mixture was pumped into the homogenizer and homogenized under 100 MPa for 1, 2, 10, 30 min, respectively. At the different time intervals, the suspension was allowed standing and the powder was collected by filtration. The obtained powder was washed with ethyl ether (99.5%) for 3 times. The product was finally dried under vacuum at 150 °C for 12 h.

## Data availability

The data that support the findings of this study are available within the article and supplementary information files, or available from the corresponding authors on request.

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

## Acknowledgements

The authors acknowledge National Natural Science Foundation of China (NO. 21978138, 31700514), the Fundamental Research Funds for the Central Universities (Nankai University), the China Postdoctoral Science Foundation (NO. 2022M721702), the Natural Science Foundation of Tianjin, China (No. 18JCYBJC86500), and the Haihe Laboratory of Sustainable Chemical Transformations (YYJC202101) for financial support of this work. Partial support from the Researchers Supporting Program (RSP2023R/55) at King Saud University, Riyadh, Saudi Arabia and the Robert A. Welch Foundation (B-0027) is also acknowledged.

## Author contributions

B.L., Y.W., and S.M. conceived and designed the research. X.L and B.L. co-wrote the manuscript. X.L., A.W., C.W., Y.W., and B.L. carried out the materials design, synthesis and characterization as well as performed synthesis experiments and data analysis. J.L., Z.Z., F.S., and Z.Y. contributed to materials characterization. A.M.A. and A.N. assisted to revise the manuscript. All authors discussed the results and commented on the manuscript.

## Competing interests

The authors have been applied by Nankai University with B.L. and X.L. as named inventors. The patent authorization number and patent application number are ZL202110257296.5, ZL20211074628.2, and CN202210127476.6, respectively. The remaining authors declare no competing interests.
