## [Peer review file · Nature Communications]

REVIEWER COMMENTS

Reviewer #1 (Remarks to the Author):

This manuscript describes the use of high-pressure homogenization (HPH) to synthesize porous materials. They synthesize 10 materials (4 phloroglucinol based COFs, 4 MOFs, and 2 POCs) at about gram scale and present a thorough screen of the optimal homogenization time for each material using PXRD, IR, BET, and the reported yield to characterize the material at each time point. They also provide TGA and stability tests (acid and base stability for the COFs but only solvent stability for the MOFs and POCs) to show the HPH synthesized materials display similar stability to other reported methods. They demonstrate scalability by synthesizing one material of each type on large scale (>50 g) and they provide detailed characterization of the large-scale sample. They also show that the solvent is recyclable by synthesizing 12 consecutive batches of a material using the same solvent (they add a little more solvent after 6 batches) that all have comparable crystallinity and surface area. I think the experiments conducted and characterization provided are good. This work will be of interest to those interested in synthesizing porous materials on large scale. However, I have a few concerns I would like to see addressed before I recommend this work for publication in Nature Communications.

Major concerns:

- 1) The MOFs the authors selected are not the most difficult to synthesize. In each case, crystalline and porous material is observed after just one minute of homogenization (with 2 min giving the highest BET surface area). Three of the MOFs (HKUST-1, ZIF-8, and ZIF-67) have "stir at room-temperature" syntheses reported that are even cited in the SI in a comparison of BETs and production rates for reported methods. I think it would be beneficial for the authors to do a control experiment and that check the PXRD and BET surface area of each of the investigated materials after just regular mixing for the same amount of time as the optimum homogenization time. This is needed to demonstrate that HPH is playing a role in accelerating the synthesis of these materials.
- 2) While all of the MIL-53-NH₂ (Al) samples are crystalline, not a single one is a good match for the simulated pattern they provide (Figure 2h). I am not sure if this is because of the flexibility of this MOF or the formation of a different phase, but the authors need to do more work to confirm that they did indeed form MIL-53-NH₂ (Al) under these conditions.

Minor comments:

- 1) The authors should include a brief description of the HPH equipment itself to the introduction. Is it expensive? Needs to be home built? I think a major component of how useful this work is to the larger community is how easy it is to get hands on a HPH set-up.
- 2) The authors claim in the intro that "the tech of large-scale industrial synthesis of these materials have not been achieved," yet two of the MOFs they demonstrate are produced commercially by BASF. They should make this clearer in the manuscript.
- 3) There is a mistake in Figure 6b. The solvothermal BET for ZIF-8 in their comparison bar chart is incorrect for the reference they provide. Based on table S2 that compares reported methods for some of the materials, I think they included the BET of the room-temperature synthesis ZIF-8 and not of the solvothermal ZIF-8 sample. This should be corrected.
- 4) The authors use close to stoichiometric ratios for most of the materials, except for ZIF-8 and ZIF-67, where a 35 to 40 fold excess of linker is used. This isn't commented on in the main text anywhere, and I think the authors should explain why or show why the excess of linker is necessary for these two materials
- 5) The SEMs of materials prepared by HPH should be carefully compared to materials prepared by other traditional methods.

Reviewer #2 (Remarks to the Author):

Ma and coworkers developed for the first time a novel and general methodology of high pressure homogenization (HPH) to realize the continuously large-scale synthesis of crystalline porous materials with high performance and high efficiency under benign conditions. The HPH method can not only overcome shortcomings of low yield, high energy consumption, low efficiency, and sophisticated preparation process for conventional methods such as hydrothermal/solvothermal, ionothermal, microwave, sonochemical synthesis and so on, but also circumvent the intrinsic drawbacks of poor mass/thermal transfer and poor reproducibility for reported MC and TSE methods. The report of this technology will greatly promote the industrial synthesis and application of crystalline porous materials. Therefore, I strongly recommend this work to be published in Nature Communications after a minor revision with the following comments addressed.

1. In the high pressure homogenization (HPH) strategy, pressure is an important parameter for synthesis. I suggest comparing different pressures in the preparation of crystalline porous materials.
2. In Fig. 4a, hysteresis was observed for the N₂ sorption isotherms, which should be explained.
3. Please add the reaction diagram of the 1, 3, 5-Triformylphloroglucinol and C4RACHO in supporting information.
4. The pore size distribution of the COFs should be provided. A narrow pore size distribution is generally expected for COFs with high crystallinity.
5. Authors mention color changes during synthesis. How do the solid-state UV-visible spectra change during synthesis and between methods?
6. In the results and discussion (line 95); "To illustrate our strategy, we selected 4 COFs (TpPa-1, TpPa-2, TpBD, DAAQ)"; looks something is wrong with this sentence, please double check.

Reviewer #3 (Remarks to the Author):

Liu and coworkers report the production of a range of porous materials through the use of homogenisation. A range of MOFs, Cos and porous cages have been synthesised by this method. The work is large in volume, thorough, and appears to be been conducted quite carefully to produce reliable results. The authors should be commended on the level of effort placed into this work.

In this reviewer's opinion the work could be suitable for publication with little to no change, but it lacks the novelty that might be expected of the field at this stage of development. It has now been around ten years since the first reports of scale up synthesis of MOFs and related materials. There are now several well-established techniques that each claim versatility, high space time yields and high production rates, similarly to this paper.

The key challenges for the field are now downstream, in the work up, shaping and sustainable production. These are the areas that should feature in reports in journals that focus upon novelty. This work is suited to a more specialised journal.

Key points to consider include:

1. The report still utilises organic solvents for the production, key to the field is utilising aqueous conditions where possible. Given the energetics of the homogenisation process, efforts to convert a solvent-based process over to aqueous conditions would be preferable.
2. The paper claims that this process lowers energy consumption. Evidence should be provided against this claim so that it can be verified by the community.
3. The work up that is reported is largely at too small a scale for the relevant issues encountered during filtration and washing to be observed. Work up research needs to be done at a much larger scale.

We really appreciate the comments and suggestions from the reviewers, and we have revised our manuscript accordingly as detailed in the responses below. The corresponding changes have been highlighted in yellow in the main text and Supplementary Information.

Reviewers' comments:

Reviewer: 1

Comment 1: The MOFs the authors selected are not the most difficult to synthesize. In each case, crystalline and porous material is observed after just one minute of homogenization (with 2 min giving the highest BET surface area). Three of the MOFs (HKUST-1, ZIF-8, and ZIF-67) have "stir at room-temperature" syntheses reported that are even cited in the SI in a comparison of BETs and production rates for reported methods. I think it would be beneficial for the authors to do a control experiment and that check the PXRD and BET surface area of each of the investigated materials after just regular mixing for the same amount of time as the optimum homogenization time. This is needed to demonstrate that HPH is playing a role in accelerating the synthesis of these materials.

Response 1: We appreciate the constructive suggestion from the reviewer. Per the suggestion, we conducted the control experiment using conventional room temperature synthesis under the same reaction time. The results showed that regular "stir at room-temperature" syntheses cannot produce most crystalline porous materials including COFs (TpPa-1, TpPa-2, TpBD, DAAQ, Supplementary Figs 24-27, Supplementary Table 1), and POCs (CPOC-301, and CC3R-OH, Supplementary Figs 63-64, Supplementary Table 1). And "stir at room-temperature" syntheses can only afford MOFs with poor crystallinity and low surface area under the same reaction time (Supplementary Figs. 47-50, Supplementary Table 1). These results thus demonstrate that HPH plays a key role in accelerating the synthesis of these materials.

Comment 2: While all of the MIL-53-NH₂ (Al) samples are crystalline, not a single one is a good match for the simulated pattern they provide (Figure 2h). I am not sure if this is because of the flexibility of this MOF or the formation of a different phase, but the authors need to do more work to confirm that they did indeed form MIL-53-NH₂ (Al) under these conditions.

Response 2: Thanks for the suggestion. The successful formation of MIL-53-NH₂ (Al) by HPH method was confirmed by the consistent PXRD patterns with those obtained via solvothermal method (*J. Catal.* 2009, 261, 75-87; *J. Am. Chem. Soc.* 2009, 131, 6326–6327). Please see Supplementary Fig. 40 for details.

Comment 3: The authors should include a brief description of the HPH equipment itself to the introduction. Is it expensive? Needs to be home built? I think a major component of how useful this work is to the larger community is how easy it is to get hands on a HPH set-up.

Response 3: We appreciate the valuable suggestion from the reviewer. Per the suggestion, we have added a brief description of the HPH equipment in the introduction, and it is a commercially available instrument with low-cost (please see page 4 in the manuscript for details).

Comment 4: The authors claim in the intro that "the tech of large-scale industrial synthesis of these materials have not been achieved," yet two of the MOFs they demonstrate are produced commercially by BASF. They should make this clearer in the manuscript.

Response 4: We have revised the corresponding description in the introduction section of the manuscript according to the suggestion from the reviewer (please see page 3 for details).

Comment 5: There is a mistake in Figure 6b. The solvothermal BET for ZIF-8 in their comparison bar chart is incorrect for the reference they provide. Based on table S2 that compares reported methods for some of the materials, I think they included the BET of the room-temperature synthesis ZIF-8 and not of the solvothermal ZIF-8 sample. This should be corrected.

Response 5: Thanks for pointing this out. We have corrected the corresponding error in the revised manuscript (please see Fig. 6b in the manuscript for details).

Comment 6: The authors use close to stoichiometric ratios for most of the materials, except for ZIF-8 and ZIF-67, where a 35 to 40 fold excess of linker is used. This isn't commented on in the main text anywhere, and I think the authors should explain why or show why the excess of linker is necessary for these two materials.

Response 6: We have carried out the experiments to optimize stoichiometric ratio for the preparation of ZIF-8 and ZIF-67. The results showed that the optimum stoichiometric ratios of Zn^{2+} :2-methylimidazole for ZIF-8 and Co^{2+} :2-methylimidazole for ZIF-67 are 1:40 and 1:35, respectively (Supplementary Figs. 41-42). We have added the relevant discussion in the revised manuscript (please see page 12 in the manuscript for details).

Comment 7: The SEMs of materials prepared by HPH should be carefully compared to materials prepared by other traditional methods.

Response 7: We have added the relevant SEMs of materials obtained via solvothermal method according to the suggestion from the reviewer (please see page 7 in the manuscript and Supplementary Figs. 13-14 for details).

Reviewer: 2

Comment 1: In the high pressure homogenization (HPH) strategy, pressure is an important parameter for synthesis. I suggest comparing different pressures in the preparation of crystalline porous materials.

Response 1: We appreciate the constructive suggestion from the reviewer. Per the suggestion, we have optimized the corresponding pressure parameters (please see page 9, 12, 15 in the manuscript and Supplementary Figs. 20-23, 43-46, and 61-62 for details).

Comment 2: In Fig. 4a, hysteresis was observed for the N₂ sorption isotherms, which should be explained.

Response 2: We have added the explanation for the hysteresis of the N₂ sorption isotherms in the revised manuscript (please see page 9 in the manuscript for details).

Comment 3: Please add the reaction diagram of the 1, 3, 5-Triformylphloroglucinol and C₄RACHO in supporting information.

Response 3: We have added the reaction diagram of the 1, 3, 5-triformylphloroglucinol and C₄RACHO in supporting information according to the suggestion from the reviewer (Please see in Page S4 in the supporting information for details).

Comment 4: The pore size distribution of the COFs should be provided. A narrow pore size distribution is generally expected for COFs with high crystallinity.

Response 4: We have added the pore size distribution of the COFs in revised supporting information (please see page 9 and 10, Supplementary Fig. 28 for details).

Comment 5: Authors mention color changes during synthesis. How do the solid-state UV-visible spectra change during synthesis and between methods?

Response 5: We have added the corresponding solid-state UV-visible spectra data in revised supporting information and discussed in the manuscript according to the suggestion from the reviewer (please see page 6, Supplementary Figs. 9-11 for details).

Comment 6: In the results and discussion (line 95); “To illustrate our strategy, we selected 4 COFs (TpPa-1, TpPa-2, TpBD, DAAQ)”; looks something is wrong with this sentence, please double check.

Response 6: Thanks for pointing this out and we have revised the corresponding description (please see page 6 in the manuscript for details).

Reviewer: 3

Comment 1: The report still utilises organic solvents for the production, key to the field is utilizing aqueous conditions where possible. Given the energetics of the homogenization process, efforts to convert a solvent-based process over to aqueous conditions would be preferable.

Response 1: We appreciate the constructive suggestion from the reviewer. Per the suggestion, we carried out the experiment by using water as solvent for the production of HKUST-1, ZIF-67, CPOC-301, and CC3R-OH based on HPH approach. As a result, HKUST-1, CPOC-301 and CC3R-OH cannot be formed in water by HPH approach, which could be presumably due to the poor solubility of organic ligands in water (Supplementary Fig. 51, Figs. 65-66, Supplementary Table 3). This may be the reason traditional solvothermal synthesis of HKUST-1, CPOC-301 and CC3R-OH also employed organic solvents as the reaction media (*J. Am. Chem. Soc.* 2006, 128, 1304-1315, *J. Am. Chem. Soc.* 2020, 142, 18060-18072, *ACS Appl. Nano Mater.* 2020, 3, 479-485). In addition, even ZIF-67 could be synthesized in water by HPH method, the optimized BET surface area (BET = 613 m² g⁻¹) of ZIF-67 using water as solvent is much lower than that of using methanol as solvent (BET = 1282 m² g⁻¹) (Supplementary Fig. 52). These results indicate that the presence of organic solvents is

beneficial for forming high quality products in some cases. The corresponding discussions have been added in the revised manuscript (Page 13, 15, and 16).

Comment 2: The paper claims that this process lowers energy consumption. Evidence should be provided against this claim so that it can be verified by the community.

Response 2: The energy consumption of HPH approach for producing different crystalline porous materials was measured by electric power monitor, which illustrates a 22-1178 times lower energy consumption compared with the solvothermal approach (please see page 10, 13, 14, 16 in the manuscript and Supplementary Figs. 32, 55, 69 for details).

Comment 3: The work up that is reported is largely at too small a scale for the relevant issues encountered during filtration and washing to be observed. Work up research needs to be done at a much larger scale.

Response 3: Thanks for the suggestion. To illustrate the feasibility of our approach for a large-scale synthesis, we carried out a kilogram-scale synthetic experiment to illustrate the practicable process when facing some issues such as filtration and washing. The results showed that about 1 kg HKUST-1 can be easily obtained via the procedures of sedimentation and centrifugal separation (please see page S7, Supplementary Fig. 73 for details).

Again, we thank the reviewers for the constructive comments and suggestions, which have made our manuscript much improved.

REVIEWERS' COMMENTS

Reviewer #1 (Remarks to the Author):

The authors have done a great job of addressing all reviewer comments. The demonstration of a kg scale MOF synthesis is impressive. This work can be published without any further revisions.

Reviewer #2 (Remarks to the Author):

The authors have thoroughly addressed all my comments, and the overall quality of the manuscript is enhanced and improved. In my opinion, I recommend this manuscript to be published in Nature Communications.

At the same time, I was asked by the editor to look over the comments of Reviewer #3. After reading the manuscript and responses carefully, I find that the authors have made significant changes to this manuscript that satisfy Reviewer #3 comments. Therefore, I think this work deserves publication now and I am happy to support the publication of this manuscript in Nature Communications.

We really appreciate the great support from both reviewers.

Reviewer #1 (Remarks to the Author):

The authors have done a great job of addressing all reviewer comments. The demonstration of a kg scale MOF synthesis is impressive. This work can be published without any further revisions.

Response: We highly appreciate the positive evaluation from the reviewer for our work.

Reviewer #2 (Remarks to the Author):

The authors have thoroughly addressed all my comments, and the overall quality of the manuscript is enhanced and improved. In my opinion, I recommend this manuscript to be published in Nature Communications. At the same time, I was asked by the editor to look over the comments of Reviewer #3. After reading the manuscript and responses carefully, I find that the authors have made significant changes to this manuscript that satisfy Reviewer #3 comments. Therefore, I think this work deserves publication now and I am happy to support the publication of this manuscript in Nature Communications.

Response: We highly appreciate the great support from the reviewer for our work.